# Standard Deviation Quantitative Characterization and Process Optimization of the Pyramidal Texture of Monocrystalline Silicon Cells

**DOI:** 10.3390/ma13030564

**Published:** 2020-01-24

**Authors:** Zheng Fang, Zhilong Xu, Tao Jang, Fei Zhou, Shixiang Huang

**Affiliations:** 1College of Mechanical and Energy Engineering, Jimei University, Xiamen 361021, China; fzheng1995@163.com (Z.F.); 17859793951@163.com (T.J.); zf18850043071@163.com (F.Z.); 2Institute of Manufacturing Engineering, Huaqiao University, Xiamen 361021, China; 17359897991@163.com

**Keywords:** pyramidal texture, relative standard deviation, quantitative characterization, silicon cell, photo-electric characteristics, process optimization

## Abstract

To quantitatively characterize the pyramidal texture of monocrystalline silicon cells and to optimize the parameters of the texturing process, the relative standard deviation *S*_h_ was proposed to quantitatively characterize the uniformity of the pyramidal texture. Referring to the definition and calculation of the standard deviation in mathematical statistics, *S*_h_ was defined as the standard deviation of the pyramid relative height h_i_ after normalization of the pyramid height H_i_ of monocrystalline silicon wafer surfaces. Six different silicon cells, with different pyramidal textures, were obtained by applying different texturing times. The relationships between *S*_h_ and the photoelectric characteristics were analyzed. The feasibility of quantitatively characterizing the uniformity of the pyramidal texture using *S*_h_ was verified. By fitting the *S*_h_ curve, the feasibility of optimizing the texturing process parameters and predicting the photoelectric characteristics using *S*_h_ was verified. The experimental and analytical results indicate that, when the relative standard deviation *S*_h_ was smaller, the uniformity of the pyramidal texture obtained by texturing was better. The photoelectric conversion efficiency (PCE) of the silicon cells monotonically increased with decreasing *S*_h_. The silicon cell obtained by texturing with 2% tetramethylammonium hydroxide (TMAH) solution for 18.1 min had a textured surface with a minimum of *S*_h_, the reflectivity of the silicon cell reached its minimum value of 2.28%, and the PCE reached its maximum value of 19.76%.

## 1. Introduction

Silicon cells occupy approximately 90% of the worldwide market share in the photovoltaic (PV) industry, due to their low cost and high photoelectric conversion efficiency (PCE) [1]. The PCE of a silicon cell can be improved in two basic ways: (1) Increasing its sunlight absorption rate and (2) enhancing its PV effect [2,3]. A pyramidal trapping texture is directly prepared on the surface of a silicon wafer by chemical texturing, which can effectively improve the sunlight absorption rate, thereby improving the PCE [4,5,6,7].

Many reports on the influence of surface texture on the photoelectric properties of silicon cells have been presented. Through three-dimensional (3D) modeling and optical simulation analysis, Moroz et al. determined that a silicon cell with a random pyramidal texture of uniform height had a lower reflectivity than a silicon cell with a random pyramidal texture of different heights [8]. Kim et al. studied the diffusion depth of different pyramidal textures and found that, compared to the other pyramidal textures, the uniform small pyramidal structure had more uniform diffusion, and the recombination rate of the carrier in the diffusion layer was lower, thus obtaining better electrical properties [9]. Dai et al. studied SiN_X_ thin film deposition on the surface of a silicon cell and found that the thickness of the SiN_X_ thin film deposited on uniform small pyramids was more uniform than that deposited on other pyramids, and its light trapping and passivation effects were better [10]. Khanna et al. studied the influence of random pyramids on the formation of silk-screen-printed silver contacts and found that a larger difference in pyramid height led to a higher contact resistance [11]. A large number of studies have shown that the better the pyramidal texture uniformity is, the better the photoelectric characteristics of the silicon cell are.

To fabricate a pyramidal texture with good uniformity, many scholars have performed a lot of research on the texturing process. Chen et al. used an alkali solution for polishing before texturing to obtain a relatively uniform distribution of a pyramidal texture on a surface, and the light absorption performance and minority carrier lifetime were significantly improved [12]. Wang et al. used a mixture of TMAH and IPA to perform chemical texturing on a silicon wafer and obtained a small and uniform pyramidal texture on the surface, which greatly improved the PCE of the silicon cell [13]. Huang et al. obtained a pyramidal texture with a high coverage rate and good uniformity by applying mask etching on a silicon nitride layer [14]. A large number of studies show that the pyramidal texture uniformity can be improved by adjusting the type of etching solution, solution concentration, additive, and etching temperature and time, which will further improve the photoelectric characteristics of the silicon cells [15,16,17,18].

The uniformity of the pyramidal texture of a silicon cell has a great influence on its photoelectric characteristics. How to characterize the uniformity of the pyramidal texture is key to regulating the chemical production process and improving the PCE. At present, most researcher determined the texture quality by combining the photoelectric characteristics and qualitatively analyzing the uniformity after observing the pyramidal surface of a silicon wafer via scanning electron microscopy (SEM) [19,20,21,22]. Some scholars have also used statistical methods to characterize the uniformity of the pyramidal texture [23,24]. Lien et al. characterized the distribution of the pyramid size on the surface of a silicon wafer by studying the proportions of pyramid sizes [25]. Wefringhaus obtained 3D surface contours by laser confocal microscopy and converted the 3D surface contours to a distribution histogram of the pyramid height and bottom length using an algorithm written in Mountains Map software [26]. While these methods can also express the distribution of the pyramids, the uniformity of the pyramidal texture cannot be accurately quantified. Combining characterization means with the photoelectric characteristics to optimize the texturing process is difficult.

To quantitatively characterize the pyramidal texture and optimize the parameters of the texturing process, this study proposed a method to quantify the uniformity of the pyramidal texture on the surface of a silicon cell by using *S*_h_ and optimized the parameters of the texturing process by using a characterization method. Different pyramidal textures on the surfaces of the wafers were obtained by applying different chemical texturing times. The feasibility of the proposed characterization method for quantitatively characterizing the pyramidal texture was verified. The relations between the quantitative characterization parameter and the photoelectric characteristics of the silicon cell were established. A parameter of the texturing process was optimized using the quantitative characterization parameter, and the feasibility was verified.

## 2. Texture Growth and Characterization

### 2.1. Texture Growth Process on the Surface of the Wafers

The periodicity and density of the atomic configuration in different directions of the crystal lattice are different, which results in different physical and chemical properties of the crystal in different directions. This anisotropy causes the (100) surface of a wafer to etch approximately 10 times faster than the (111) surface in alkaline conditions, resulting in a pyramidal texture being etched onto the surface [27]. The surface of a wafer can be etched into uniform quadrangular pyramids under ideal conditions. However, due to the influence of the uneven surface states of the wafer before etching, the uneven texturing solution and the different effusion rates of bubbles on the surface of the silicon wafer, the pyramids have inconsistent sizes, irregular structures, and overlapped stacking.

Three typical pyramidal textures occur under chemical texturing according to the growth characteristics of the wafer for different etching times. During the initial etching, pyramids just cover the surface of the wafer. The overall size of the pyramids are small, and many nanoscale pyramids form, resulting in a large relative size gap between the pyramids, as shown in Figure 1a. With moderate etching, the tiny pyramids gradually grow, which reduces the relative size gap between the pyramids and improves the texture uniformity, as shown in Figure 1b. Under excessive etching, the pyramid size increases, and at the same time, new small pyramids are formed; thus, the size difference of the pyramids increases, and the uniformity of the texture decreases, as shown in Figure 1c.

### 2.2. Quantitative Characterization of Pyramidal Texture

To quantify the uniformity of the pyramidal texture, the standard deviation concept from mathematical statistics was used to evaluate the pyramid height difference of the wafer. The standard deviation is the square root of the arithmetic mean of the deviation squared between individual values and their mean. The standard deviation reflects the degree of dispersion between individual values within a group. As the texturing time increases, the pyramid height on the surface of the wafer continuously increases. If the standard deviation of the pyramid height is directly used to evaluate the uniformity, then deviations of the reference heights will lead to errors. Therefore, the pyramid height of the surface texture is normalized first. SEM images of the pyramidal texture of the wafers were obtained by SEM, and the height Hi of each pyramid in the SEM images was calculated by image processing software. According to Equation (1), the average height *H*_a_ of the surface pyramids of a wafer was calculated. According to Equation (2), the pyramid relative height *h*_i_ of the wafer was calculated. According to Equation (3), the relative standard deviation *S*_h_ of the pyramidal texture on the surface of the wafer was calculated.
(1)$$H_{a}=\frac{\sum_{i=1}^{n}H_{i}}{n}$$
(2)$$h_{i}=\frac{H_{i}}{H_{a}}$$
(3)$$S_{h}=\sqrt{\frac{1}{n}\sum_{i=1}^{n}{\left(h_{i}-1\right)}^{2}}$$

*S*_h_ can be used to effectively evaluate the uniformity of the pyramidal texture on the surface of a wafer. When the pyramidal texture is more non-uniform, the deviation between the relative height hi and the mean value 1 after normalization is larger, and *S*_h_ is larger. When the pyramidal texture is more uniform, the deviation between the relative height *h*_i_ and the normalized mean value 1 is smaller, and *S*_h_ is smaller. When the pyramidal texture is ideal and completely uniform, *S*_h_ is 0. When *S*_h_ is positive and closer to 0, the uniformity of the pyramidal texture is better.

### 2.3. Calculation of Relative Standard Deviation

To obtain the pyramid heights on the surface of a wafer, SEM images of the surface morphology of the wafer were imported into the image processing software (Visual Studio). The projected lengths of the four edges of each pyramid can be automatically identified, and the random pyramidal structures on the surface of the wafer are approximately upright rectangular pyramidal structures. Each pyramid height *H*_i_ was calculated according to the average projection lengths of the four edges of the pyramid. According to the SEM images of the three typical pyramidal textures in Figure 1, the pyramid height distributions in a 40 μm × 40 μm region were obtained, as shown in Figure 2. In Figure 2a, 862 small pyramids were measured in the sampling area for initial etching; the pyramid heights are less than 1.6 μm, and most of the pyramid heights are between 0.2 μm and 0.8 μm. Substituting each pyramid height *H*_i_ into Equation (1), the average pyramid height *H*_a1_ is 0.71 μm. In Figure 2b, 619 pyramids were measured for moderate etching in the same sampling area. The pyramid heights are between 0.6 μm and 1.4 μm, and the maximum height is less than 2.4 μm. The average height *H*_a2_ was calculated to be 0.98 μm. In Figure 2c, the number of pyramids measured in the sampling area for excessive etching was reduced to 222; the pyramid height was considerably increased, and some new small pyramids also grew at the same time. The pyramids are scattered, and the height difference is widened. The average height of the pyramids *H*_a3_ was calculated to be 2.06 μm.

The pyramid heights of the three textures were substituted into Equation (2) for normalization, and the relative heights of the pyramids (*h*_i_ = *H*_i_/*H*_a_) were obtained. The mean pyramid relative height hi is 1, and the distribution histograms of hi are shown in Figure 3. Figure 3a shows the relative height distribution of the pyramids for initial etching; the relative height is within the range of 0–2.5, and the number of pyramids near the mean value 1 is the highest. The histogram roughly shows a normal distribution centered on the mean value, but a few pyramids do not conform to the above distribution. The number of pyramids with relatively small or large heights is higher than that with moderate etching. Figure 3b shows the distribution of the pyramid relative height with moderate etching. The distribution is relatively concentrated in strict accordance with the normal distribution, with a mean of 1 as the center. Figure 3c shows the distribution of the pyramid relative height with excessive etching; its distribution is more dispersed.

The pyramid relative height hi of the three textures was substituted into Equation (3) to calculate the relative standard deviation *S*_h_ of each group. The relative standard deviations of the three textures are *S*_h1_ = 0.456, *S*_h2_ = 0.419, and *S*_h3_ = 0.503. *S*_h_ first decreases and then increases by increasing the etching time; *S*_h_ is the smallest when the etching is moderate, and the uniformity of the pyramidal texture is the best.

## 3. Materials and Methods

The experiment used a P-type boron-doped (100) crystal wafer with an area of 156 × 156 mm, a thickness of approximately 180 μm and a resistivity of 0.4 to 1.5 Ω·cm. A total of 120 pieces of the wafer were polished with 30% (mass fraction) NaOH solution at 70 °C for 3 min to remove the machine-damaged layer on the surface of the wafers. Then, the polished wafers were divided into 6 groups of 20 pieces each, and a mixture of 2% (mass fraction) TMAH + 8% (volume fraction) IPA + 4% (mass fraction) Na_2_SiO_3_ was used to etch for 10, 15, 20, 25, 30, and 35 min at 80 °C. Six different pyramidal textures of the wafers were obtained.

The silicon cells were made after the wafers were textured according to the following procedures: The RCA (industrial standard wet method) cleaning process was used to clean the wafers after texturing. p-n junctions were made using a DS-300L tubular diffusion furnace at 870 °C for 60 min under POCl_3_ gas as a phosphorus source. The phosphorus-diffused wafers were immersed in a 9% (*Volume fraction*) HF solution for wet etching to remove phosphorus silica glass. SiO_2_ thin film was generated by heating oxidation of the silicon wafers in an annealing furnace with O_3_ flow. Plasma enhanced chemical vapor deposition (PECVD) was used to electroplate 70 nm thick SiNx on the front of the wafers. The silicon cells were fabricated by screen printing an electrode and sintering the samples at 650 °C.

After texturing and cleaning the wafer, the surface texture of it was obtained using scanning electron microscopy (SEM) on a Phenom XL instrument, SEM images of the surface morphology of the wafers were imported into Visual Studio image processing software to obtain the *H*_i_ of the pyramid heights of each group. After the coating process, the reflectance values for the wafers in each group was measured by a ZhiDong photoelectric D8 integral reflectometer, and the minority carrier life of the wafer in each group was measured by a Semilab WT-1000B minority carrier life tester. After sintering the cells, the current-voltage characteristics of the silicon cells were tested using a SolarIV-1000 solar cell test system.

## 4. Results and Discussion

### 4.1. Surface Morphology of the Wafers

If scanning electron microscope (SEM) observations of each group of samples is shown by multiple images, it is easy to confuse the reader and it is inconvenient to see the change law. The representative surface morphology (with a sampling area of 53.6 × 53.6 μm) of the 20 wafers are selected and shown in Figure 4. The surface texture of the wafer with a texturing time of 10 min is shown in Figure 4a; the surface is covered with small pyramids. As the texturing time increased, some of the pyramids became larger, some of the pyramids were etched away, and new pyramids grew. Figure 4b shows the surface morphology of the wafer textured for 15 min, and the number of pyramids in the same area is reduced. The surface morphology of the wafer with a texturing time of 20 min is shown in Figure 4c. The surface pyramidal size is moderate, and the textured uniformity is good. As the texturing time was extended, the pyramid sizes increased, some small pyramids were etched, and the uniformity of the pyramidal texture worsened. The texture on the surface of the wafer with a texturing time of 25 min is shown in Figure 4d. Figure 4e shows the surface morphology of the wafer textured for 30 min. The size gap of the pyramids is further increased, and large pyramids become the majority of the pyramids. Figure 4f shows the surface morphology of the wafer textured for 35 min. The number of pyramids in the same area is the smallest, the difference in the sizes of the pyramids is the largest, and the uniformity of the texture is the worst.

### 4.2. Relative Standard deviation (*S*_h_) of the Pyramidal Texture

Image processing was used to obtain the height of the pyramid texture on the surface of the wafers. To directly evaluate the height distribution of the pyramidal textures on the surfaces of the wafers, the *H*_i_ of each group was imported into Origin to obtain the distribution histogram of *H*_i_. If the corresponding height distribution diagrams of each group of samples is shown by multiple images, it is easy to confuse the reader and inconvenient to see the change law. The above representative surface morphology of the 20 wafers was selected, and the corresponding height distribution diagrams are shown in Figure 5. The height distribution of the pyramids on the surface of the wafer with a texturing time of 10 min is shown in Figure 5a. The heights of the pyramids are less than 1.6 μm, and the heights of most pyramids are concentrated between 0.2 and 0.8 μm. The number of pyramids in the sampling area reaches 918. The height distribution of the pyramids on the surface of the wafer with a texturing time of 15 min is shown in Figure 5b. The heights of the pyramids are less than 2.2 μm, and most of the pyramid heights are 0.4–1.2 μm. The height distribution of the pyramids on the surface of the wafer with a texturing time of 20 min is shown in Figure 5c. The pyramid heights are mostly between 0.6 and 1.4 μm, and the maximum pyramid height is 2.4 μm. The height distribution of the pyramids on the surface of the wafer with a texturing time of 25 min is shown in Figure 5d. The growth speed of the pyramids was accelerated, and the average height and maximum height of the pyramids were both greatly increased, with the maximum height of the pyramids reaching 3.4 μm. The height distribution of the pyramids on the surface of the wafer with a texturing time of 30 min is shown in Figure 5e; the pyramid heights are scattered and distributed between 0 and 4.4 μm. The height distribution of the pyramids on the surface of the wafer with a texturing time of 35 min is shown in Figure 5f; the number of pyramids in the sampling area is at least 231, and the pyramid height varies from 0.2 to 5.4 μm, with the largest difference.

The pyramid height distribution can only express the height distribution within the interval and cannot accurately reflect the dispersion degree of the pyramid height. First, the pyramid height *H*_i_ was substituted into Equation (1) to obtain the pyramid average height *H*_a_. Then, *H*_i_ and *H*_a_ were substituted into Equation (2) for normalization to obtain the pyramid relative height *h*i of each pyramid. Finally, the *S*_h_ of each group was obtained by substituting *h*i into Equation (3). The scattered points are shown in Figure 6, which is the test results of 20 wafers, and the broken line is the average connection line. The *S*_h_ of the pyramidal texture on the surface of the wafers first decreases and then increases with increasing texturing time. The minimum *S*_h_ is 0.422, with the best uniformity, when the texturing time is 20 min. After 30 min of texturing, the *S*_h_ growth rate obviously slows down, and the uniformity changes a little. The main reasons are as follows: The damaged layer is not completely removed to ensure the thickness of silicon wafer during the polishing of the wafer, resulting in surface grooves. At the initial etching, the etching speed and the etching order are greatly different, which leads to the relatively large size difference of the pyramid and the large *S*_h_. With the increase of etching time, the grooves on the surface of the wafer tend to be flat, and the size of the small pyramid increases gradually, resulting in the decrease of the relative size difference of the pyramid and the decrease of *S*_h_. With the further increase of etching time, the size of part of the pyramid continues to increase, part of the pyramid is etched out, and new small pyramids are formed constantly, which leads to the increase of the relative size difference of the pyramid and the gradual increase of *S*_h_. When the surface pyramidal texture is more uniform, the height distribution of the pyramids is more concentrated, and *S*_h_ is smaller. *S*_h_ can be used to quantitatively characterize the uniformity of the pyramidal texture.

### 4.3. Reflectivity of the Wafers

The reflectivity of the wafers were measured when the incident wavelength was 300–900nm, and the average value of each group of 20 pieces was calculated, as shown in Figure 7. In the early stage of texturing, the surface of the wafer is covered with small pyramids. The pyramid trapping effect is poor due to the small size, the texture is nonuniform, *S*_h_ is large, and the reflectivity is also large. As the texturing time is lengthened, the pyramid heights of the surface texture of the wafer increase, the uniformity of the texture becomes better, *S*_h_ decreases, the light trapping effect is improved, and the reflectance rapidly decreases. When the etching time is 20 min, the minimum reflectance of 2.48% is obtained. As the texturing time is further lengthened, the uniformity of the texture worsens, *S*_h_ increases, and the surface reflectance of the wafer tends to gradually increase. The reflectivity and *S*_h_ appear to first decrease and then increase with increasing texturing time.

### 4.4. Minority Carrier Lifetime of the Wafers

The minority carrier lifetime of each group of the wafers is shown in Figure 8, the minority carrier lifetime of the wafers is the longest in the early stage of texturing, and the minority carrier lifetime of the wafers gradually decreases with increasing texturing time. The main reasons are as follows: As the etching time increases, the pyramid height of the wafer gradually increases, the stacking between the pyramids is more serious, and the top angle and edge of the pyramidal texture are sharper. Due to the concentration of thermal stress, tiny cracks and defects appear in the bottom of the pyramids with the high-temperature diffusion process. The film thickness is nonuniform or the film is broken during the plating process. Moreover, cracking and fracture of the antireflection film becomes more serious with increasing pyramid height. These defects will increase the chance of minority carrier recombination on the surface of the wafer, resulting in a significant decline in the minority carrier lifetime of the wafer.

### 4.5. Electrical Properties of the Silicon Cells

As shown in Figure 9, the electrical properties of the silicon cells was measured, and the average value of 20 pieces in each test group was calculated. With increasing etching time, the open-circuit voltage *V*_OC_ and fill factor *FF* gradually decrease. The main reasons are as follows: With a longer etching time, the pyramid sizes of the silicon cell increase, and the pyramidal texture will lead to an increase in the surface defects of the silicon cell in the process of high-temperature diffusion and sintering, resulting in an increase in the reverse saturation current of the p-n junction; thus, the PV effect of the silicon cells is weakened. With increasing etching time, the short-circuit current density *J*_SC_ and PCE both first increase and then decrease. The main reasons are as follows: The pyramidal texture on the surface of the silicon cell can effectively reduce the reflection loss; when the surface is covered with a pyramidal texture and the pyramid heights are small, the increase in the light absorbance is greater than the reduction in the PV effect, so the PCE presents an upward trend. However, with further extension of the etching time, the pyramid sizes on the surface of the silicon cell further increase, the uniformity worsens, and the reflection loss increases, which leads to a decrease in the PV effect; thus, the PCE presents a rapid declining trend. A small pyramidal texture with good uniformity was obtained with a 2% TMAH solution for an etching time of 20 min, and the maximum PCE was 19.54%.

### 4.6. Relations between the Relative Standard Deviation and Photoelectric Characteristics

The uniformity of the pyramidal texture on the surface of a silicon cell has a great influence on its photoelectric characteristics. The corresponding relations between *S*_h_ and the photoelectric characteristics (reflectivity, minority carrier lifetime, *V*_OC_, *J*_SC_, *FF*, and PCE) were analyzed, as shown in Figure 10. With the gradual increase of *S*_h_, the reflectivity of the silicon cell generally shows a gradual increasing trend. When the pyramids are uniform, light is reflected by the adjacent pyramids and absorbed by the silicon cell, which enhances the light trapping effect. With the minority carrier lifetime, *V*_OC_ and *FF* greatly fluctuate with increasing *S*_h_; the longer the etching time is, the larger the pyramid size will be, resulting in more defects on the surface of the wafer and more carrier recombination, which is unrelated to the uniformity of the pyramids. *J*_SC_ and PCE tend to gradually decrease with increasing *S*_h_; the smaller the *S*_h_ is, the larger the *J*_SC_ and PCE are. The minimum value of *S*_h_ is 0.422, and the corresponding PCE reaches a maximum value of 19.54%.

According to the above experimental analysis, *S*_h_ is significantly correlated with pyramidal uniformity, reflectivity, *J*_SC_, and the PCE of a silicon cell. *S*_h_ can be used not only to evaluate the quality of the silicon cell texturing process, but also to indirectly evaluate the photoelectric performance without a complete silicon cell preparation process.

### 4.7. Optimization of Process Parameters

According to the relations between the relative standard deviation *S*_h_ of the surface pyramidal texture of silicon wafer and the photoelectric characteristics, the smaller the *S*_h_ is, the better the photoelectric characteristics of the silicon wafer are. As shown in Figure 11a, according to the relation between the texturing time and the mean of the *S*_h_, the minimum value of the *S*_h_ curve can be obtained through polynomial fitting analysis in Origin; when the fitting is a fifth degree polynomial, the fitting deviation is less than 10^−6^. When the texturing time is 18.1 min, the minimum relative standard deviation *S*_h_ obtained by the curve fitting is 0.417.

According to the optimization result of the *S*_h_ curve fitting, the 20 wafers were etched for 18.1 min under the original texturing experiment conditions. The surface morphology of the 20 wafers were measured, and the pyramidal height of 20 samples was obtained through image processing. According to Equations (1)–(3), the average *S*_h_ of the surface pyramidal texture of 20 samples is 0.416, which is close to the lowest value of the fitted *S*_h_ curve. The mean reflectivity is shown in Figure 11b.

The current-voltage characteristic test system was used to measure the electrical properties of the 20 wafers with texturing times of 20 min and 18.1 min. As shown in Figure 12a, the *J*_SC_, *V*_OC_, and *FF* of the silicon cells are all improved to different degrees after the process optimization. The mean of PCE is 19.76%, which is 0.22% higher than that of the silicon cell textured for 20 min. As shown in Figure 12b, the *J-V* curve of the best-efficiency solar cell after process optimization was measured, the *V*_OC_ is 632.44mV, the *J*_SC_ is 39.47mA/cm^2^, and the PCE has the best efficiency of 19.89%. The *S*_h_ of the optimized silicon cell is the smallest, and the PCE reaches the maximum value after optimizing the texturing time. Therefore, *S*_h_ can be used to quantitatively characterize the uniformity of the pyramidal texture, optimize the texturing process parameters, and predict the photoelectric characteristics in the texturing process of a silicon cell.

## 5. Conclusions

Quantitative characterizations of the uniformity of the pyramidal texture on the surface of a silicon cell and the optimization of the texturing process have been studied. The relative standard deviation *S*_h_ has been proposed. Its definition, calculation, experimental verification, and use in process optimization have been presented.

(1) Referring to the definition and calculation of the standard deviation in mathematical statistics, *S*_h_ was defined as the standard deviation of the pyramid relative height h_i_ after normalization of the pyramid height H_i_ of a wafer. The pyramid height H_i_ was obtained by image processing of SEM images of the surface of the wafer, and *S*_h_ was conveniently calculated.

(2) Different process parameters were used for chemical texturing of the wafers, and *S*_h_ was used to quantitatively characterize the uniformity of the pyramidal textures. The experimental results indicated that better uniformity of the pyramidal texture led to a smaller *S*_h_ and a higher PCE. *S*_h_ can be used to effectively evaluate the quality of the texturing process and indirectly evaluate the photoelectric characteristics of a silicon cell.

(3) *S*_h_ fitting was employed for process optimization. When the wafers were textured with 2% TMAH solution for 18.1 min, the *S*_h_ of the pyramidal texture of the silicon cells reache a minimum of 0.416, while the reflectivity reaches a minimum of 2.28%. Moreover, the mean of PCE reaches a maximum of 19.76%. The feasibility of optimizing the process parameters and predicting the photoelectric characteristics using *S*_h_ was verified.

## Figures and Tables

**Figure 1 materials-13-00564-f001:**

Three typical pyramidal textures obtained by a chemical etching process: (**a**) Initial etching, (**b**) moderate etching, (**c**) excessive etching.

**Figure 2 materials-13-00564-f002:**
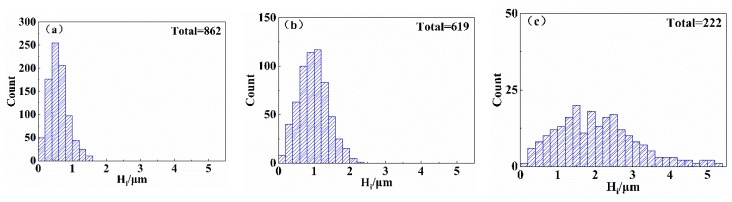
Histogram of height distribution of the three typical pyramidal textures: (**a**) Initial etching, (**b**) moderate etching, (**c**) excessive etching.

**Figure 3 materials-13-00564-f003:**
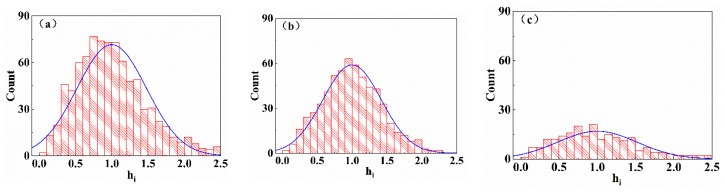
Relative height distributions of the three typical pyramidal textures: (**a**) Initial etching, (**b**) moderate etching, (**c**) excessive etching (the relative height is within the range of 0–2.5, and the number of pyramids near the mean value 1 is the highest.)

**Figure 4 materials-13-00564-f004:**
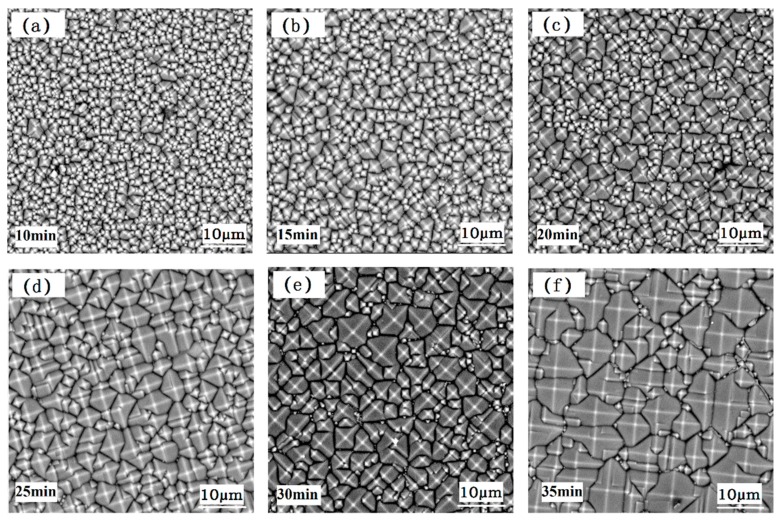
Surface morphology of the wafers: (**a**) Etching for 10 min, (**b**) etching for 15 min, (**c**) etching for 20 min, (**d**) etching for 25 min, (**e**) etching for 30 min, and (**f**) etching for 35 min.

**Figure 5 materials-13-00564-f005:**
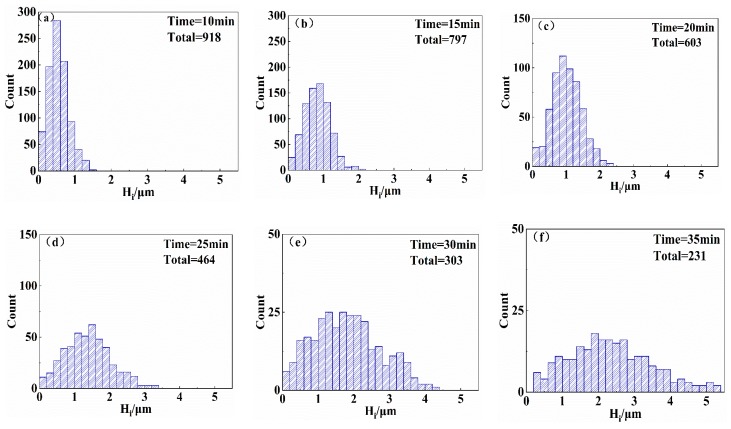
Height distributions of pyramids on the surfaces of the wafers: (**a**) Etching for 10 min, (**b**) etching for 15 min, (**c**) etching for 20 min, (**d**) etching for 25 min, (**e**) etching for 30 min, and (**f**) etching for 35 min

**Figure 6 materials-13-00564-f006:**
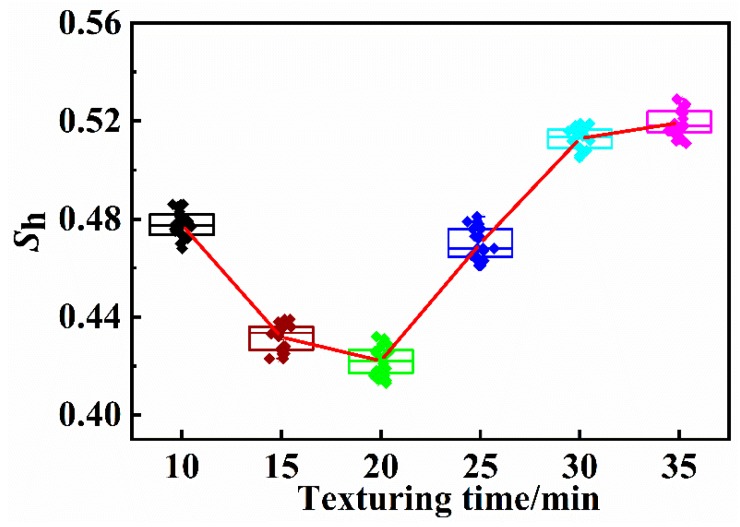
Relation between the relative standard deviation *S*_h_ and texturing time.( the *S*_h_ of the pyramidal texture on the surface of the wafers first decreases and then increases with increasing texturing time.)

**Figure 7 materials-13-00564-f007:**
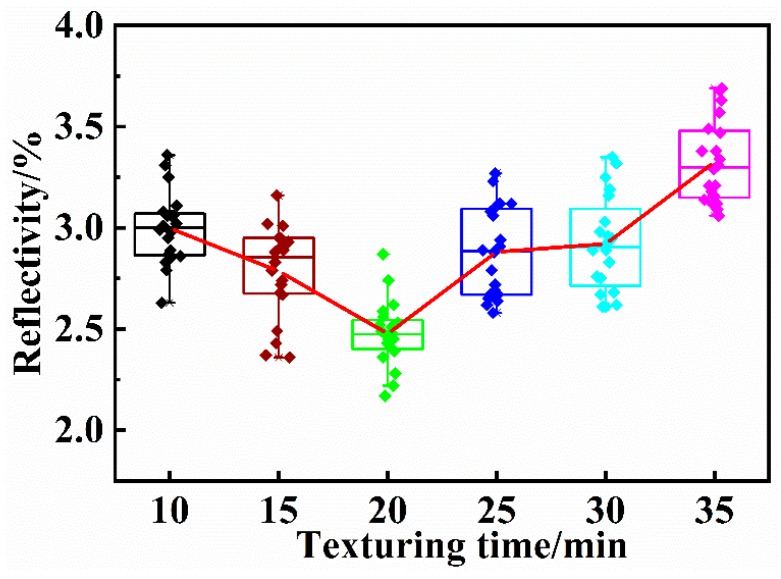
Reflectivity of the wafers. (the reflectivity and *S*_h_ appear to first decrease and then increase with increasing texturing time.)

**Figure 8 materials-13-00564-f008:**
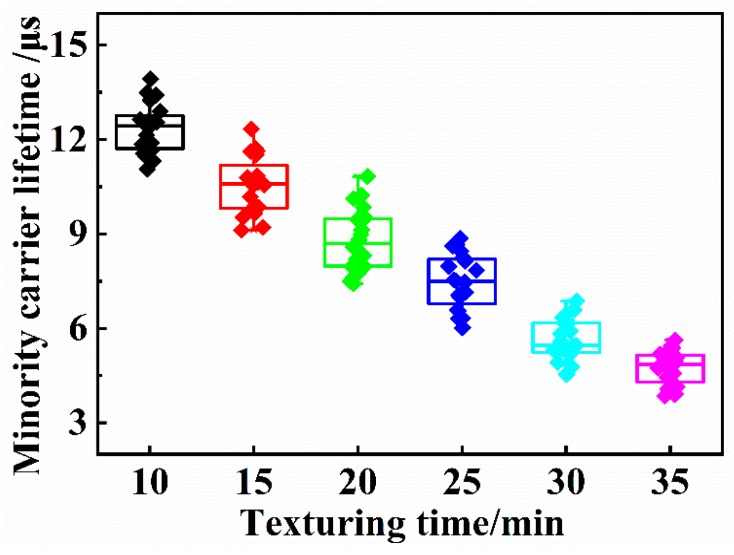
Minority carrier lifetimes of the wafers. (the minority carrier lifetime of the wafers gradually decreases with increasing texturing time.)

**Figure 9 materials-13-00564-f009:**
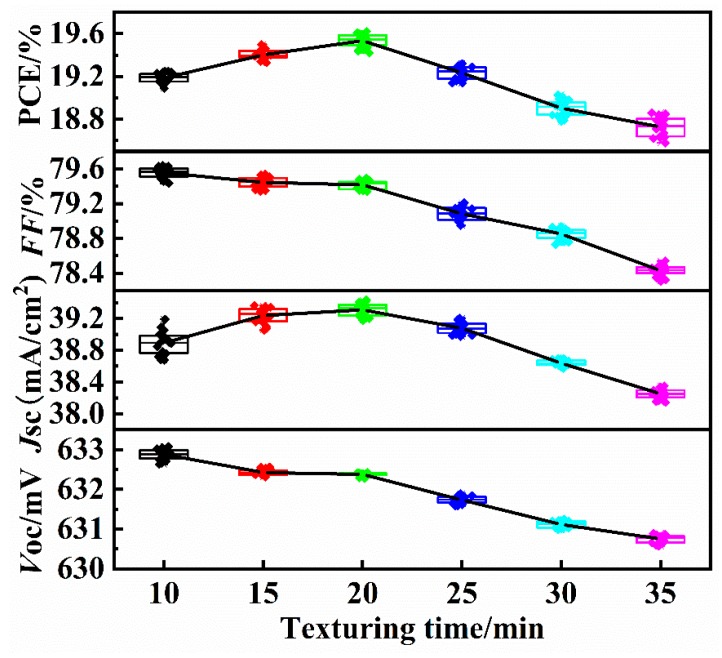
Photoelectric performance of the silicon cells. (with increasing etching time, the open-circuit voltage *V*_OC_ and fill factor *FF* gradually decrease, the short-circuit current density *J*_SC_ and PCE both first increase and then decrease.)

**Figure 10 materials-13-00564-f010:**
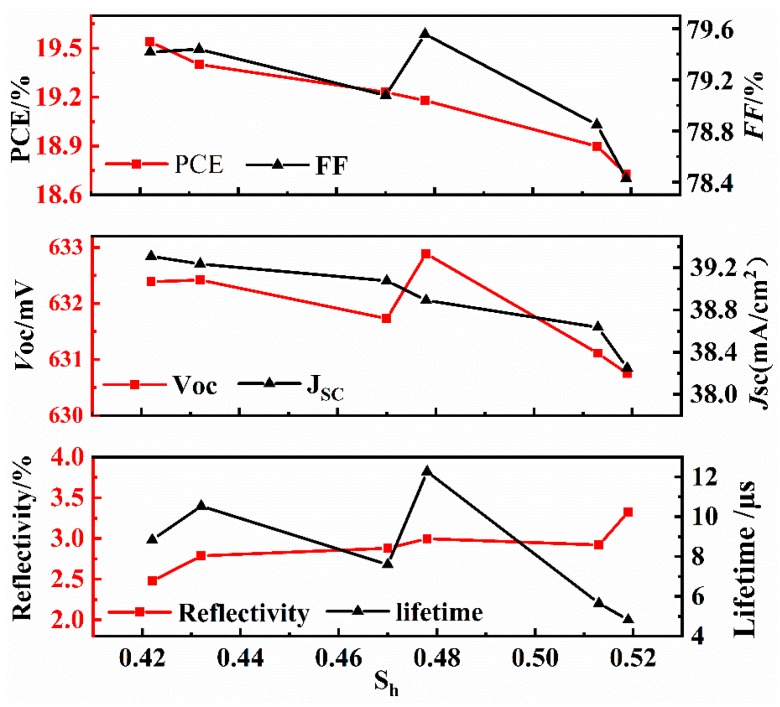
Relations between the relative standard deviation and photoelectric characteristics.

**Figure 11 materials-13-00564-f011:**
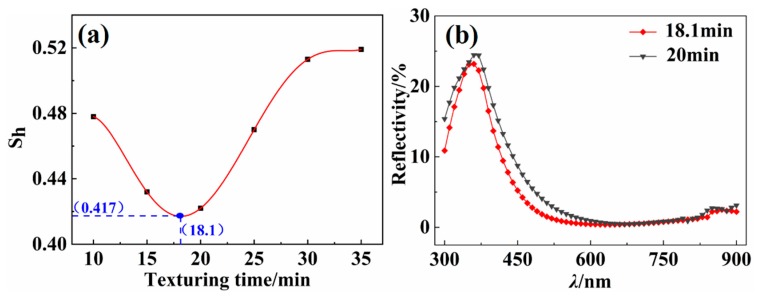
(**a**) A polynomial fitting by the relationship curve between the texturing time and *S*_h_. (**b**) The reflectivity of the silicon cell before and after process optimization of each group of 20 wafers is calculated, the reflectivity of the silicon cell textured for 18.1 min is significantly lower than that of the silicon cell textured for 20 min at the short wavelengths (λ) of 300–600 nm. The weighted reflectance of the optimized silicon cell is 2.28% at the full wavelength, which is 0.20% lower than that of the silicon cell textured for 20 min.

**Figure 12 materials-13-00564-f012:**
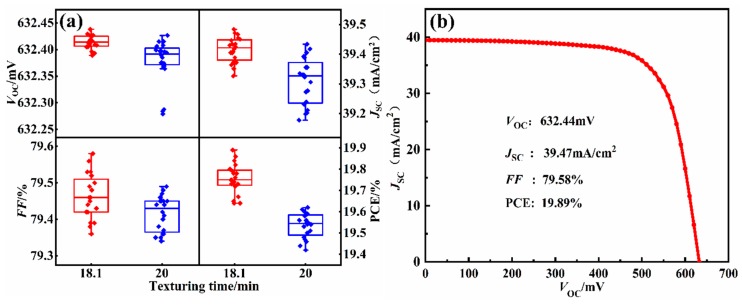
(**a**) Electrical properties of the silicon cells before and after process optimization. (**b**) The J-V curve of the best-efficiency solar cell after process optimization. (the *J*_SC_, *V*_OC_, and *FF* of the silicon cells are all improved to different degrees after the process optimization. The mean of PCE is 19.76%, which is 0.22% higher than that of the silicon cell textured for 20 min. The *J-V* curve of the best-efficiency solar cell after process optimization was measured, the *V*_OC_ is 632.44mV, the *J*_SC_ is 39.47mA/cm^2^, and the PCE has the best efficiency of 19.89%.)

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
