# Peer review of "Standard Deviation Quantitative Characterization and Process Optimization of the Pyramidal Texture of Monocrystalline Silicon Cells"

_materials, 2020, doi:10.3390/ma13030564_

Round 1

Reviewer 1 Report

The the authors introduce a statistical parameter to characterize the optical and electrical properties of surface textured solar cells made from mono-crystalline silicon. Their findings are used to interpret the dependence of the solar cell performance on the surface texture. The subject is of high interest for the scientific community. The manuscript is well organized and the methods described are adequate.

Introducing the standard deviation of a Gauss distribution as a measure for the “quality” of the texture could be improved and discussed in more detail, because:

1) The underlying statistic does not obey a normal distribution which requires values from -infinity to +infinity. The heights distribution however is limited to a lower value of zero. The deviation of the actual distribution from a normal distribution is shown in Fig.3. As a “practical” solution to the problem however it might be a reasonable choice.

2) Any statistical result strongly depend on the sample size(s). In the present manuscript the sizes vary between 918 and 231 samples (Fig.5). It seems that in each case the ~ 50µm2 -SEM image was processed as a whole for obaining the histograms, mean values and standard deviations. It would be of great interest in how far the results deviate once the sample sizes are changed. On easy way could be to divide i.e. the image of Fig.4a in 2 or 4 subsets and evaluate the results etc. This allows you to extend Table 1 by something like an error of the standard deviation and shows how accurate the parameter is determined.

When Sh is shown in Fig.6 with “error bars” together with the experimental data of the reflectance (in optics this is the more precise term for the measurement on extended rough surfaces) measurements on 20 samples it allows the reader to judge how sensitive the statistical standard deviation responds to changes in the texture as can be seen in the reflectance results. Please insert the wavelength(s) of the incident light source In your manuscript.

I recommend to add error bars for the parameters shown in Fig. 8 in the same way as it was done in Fig.7. When I look to Figs. 7 and 8 I find it quite surprising that the rather large reduction of the minority carrier lifetime from ~12µs down to 4µs (1/3) only results in a small reduction of the PCE and Isc.

Table 2 for me is misleading and should be improved because the solar cell parameters for the case of an etch time of 20 min is the mean of 20 samples ( errors should be included). Whereas the results for the etch time of 18.1 min seems to apply for one single cell only as suggested by the plots/image of Fig.10.

Author Response

Thank you very much for your valuable time and comments, please find our reply in attachment.

Reviewer 2 Report

The author tried to understand the relation between the standard deviation of randomly textured pyramids with the solar cell performance. It seems creative to analyze and correlate those two parameters. However, detailed explanation about the experimental result is required.

- Reference check is needed. (e.g. The two references, Ref 1 and Ref 2, seem to have changed)

- For Reference 2, is there any specific reason that you chose Solar cell efficiency table version 45? The most recently released version is version 55, which was released in 2019.

- About using acronyms such as MSC and MSW, I would like to ask if solar cell researchers usually use acronyms for that words, and if these acronyms may confuse readers.

- Chemical symbol bottom-up subscript correction is required.(e.g. POCl3, SiO2, O3)

- p. 8, line 182, the word of phosphorus silicon glass seem to be scientifically unaccurate, and do you mean PSG (phosphosilicate glass or phosphorus silica glass)?

- In Fig. 5, about counting number of textures, it is suggested to specify the size of images you counted the pyramids (e.g. 2mm * 1mm)

- Is it possible to explain the cause of the decrease and increase in standard deviation according to the texture time in the context?

- I would like to suggest that descriptions of the experimental methods be included in section 2, not in the section of result and discussion. (e.g. page 5 line 188, page 6 line 206, page 7 line 244, page 7 line 257, page 8 line 272)

Round 2

Reviewer 1 Report

In line 359 Sh is given by 0.417

In line 368 the weighted reflectivity is given by 2.28%

In the conclusions at line 401 Sh and refl. differ slightly Sh=0.416 and r=2.26%

Please be consistent.

Author Response

please find our reply in attachment.

Reviewer 2 Report

The authors suggest the way to evaluate the characteristics of pyramids of silicon solar cells, which could be very useful for many solar cell researchers and engineers. And also, this study is meaningful as it suggested the method that can be used to improve solar cell efficiency according to pyramid texturing process. Thus, this study is required as an answer to the questions that solar cell can have, and also will help the improvement of the solar research field.

However, still minor parts should be improved.

Although I questioned about using the acronym and suggested using the full work in 1st review, repeated use of long words throughout the paper seems inefficient, such as "monocrystalline silicon wafers" or "monocrystalline silicon solar cells".
I would like to suggest the author use the word "the solar cells" or "the silicon cells" or "the wafers" if the author used the same wafer throughout the study.
(unless your results do not include multicrystalline silicon wafer, I think there's no need to mention "monocrystalline silicon" repeatedly. It would be good to mention the word, "monocrystalln silicon" only once in the beginning.) I would like to suggest the author include the original I-V curve graph of the best-efficiency solar cell. Fig. 9, Fig. 10, and Fig. 12 : As ISC is determined by the size of silicon solar cells, I would like to suggest the use of parameter, JSC. (Jsc is not a function of cell size, thus it would be more appropriate to use the parameter of JSC instead of ISC)

Also, I appreciate the author to improve their work by accepting the 1st review comments.
